**∂ | Open Peer Review** | Clinical Microbiology | Observation

# Surveillance for *Chlamydia trachomatis* variants escaping detection with the Aptima Combo 2 assay in Canada from 2019 to 2021

Felicia Roy,[1] Jennifer Beirnes,[1] Jason J. LeBlanc,[2,3] Suzanne Gibbons,[1] Aida Sivro,[1,4,5,6] Alberto Severini,[1,4] On behalf of the Canadian Public Health Laboratory Network (CPHLN) Chlamydia Variant Working Group

**ABSTRACT**  Nucleic acid amplification tests (NAATs) are the method of choice for *Chlamydia trachomatis* diagnosis, but these strategies are susceptible to target site mutations. *C. trachomatis* variants escaping detection with the Aptima Combo 2 (AC2) assay on the Hologic Panther instrument from 23S rRNA mutations have been reported in Nordic countries, England, Japan, and the United States. Given the potential for false negative results, this study investigated whether strains of *C. trachomatis* with AC2 target site mutations were present in Canada. Surveillance was conducted in Canadian laboratories from 2019 to 2021. Specimens suspected of AC2 target site mutations included those with low-value detections on the AC2 assay, with subsequent high-value detections on the Aptima Chlamydia Trachomatis (ACT) assay used for confirmatory testing. Specimens with AC2/ACT discrepant results were subjected to sequencing of the AC2 target (i.e., 23S rRNA). Sequencing revealed 15 (4.8%) diagnostic escape variants which were carrying either C1514T, G1523A, or G1526A mutations. All specimens with a diagnostic escape mutation were detected with a reformulated version of the AC2 assay. Overall, while the prevalence of *C. trachomatis* variants was rare, their presence in the Canadian population supports the use of the new AC2 kit formulation and the need for ongoing genetic surveillance for NAAT-based assays.

**IMPORTANCE** Molecular tests are commonly used for the detection of sexually transmitted infections (STIs) like *Chlamydia trachomatis*. Mutations impacting *C. trachomatis* molecular target detection on the Hologic Panther AC2 assay have been reported in several countries, raising concerns about potential false negative results. This study showed *C. trachomatis* target detection failures in specimens submitted for *C. trachomatis* testing in Canadian laboratories from 2019 to 2021. A reformulated version of the AC2 molecular test is now available that can identify *C. trachomatis* strains harboring target site mutations that were impacted by the previous test formulation. While target site mutations were rare in Canada, revealing their presence is important to ensure accurate molecular detection of *C. trachomatis* with existing testing methods. This study supports ongoing genetic monitoring of *C. trachomatis* molecular test target sites, as well as the use of the reformulated test to avoid false negative results and subsequent transmissions.

**KEYWORDS**  *Chlamydia*, variant, mutation, surveillance, sequencing, diagnostic, Aptima, Panther

*C*hlamydia trachomatis (CT) remains the most prevalent bacterial STI, with an estimated 130 million new infections reported globally each year (1). Clinical sequelae can include pelvic inflammatory disease, ectopic pregnancy, and infertility;

**Peer Reviewers** Collette Bromhead, Massey University, Wellington, New Zealand; Jennifer Anne Danielewski, Royal Women's Hospital, Parkville, Victoria, Australia

Address correspondence to Felicia Roy, felicia.roy@phac-aspc.gc.ca.

Felicia Roy and Jennifer Beirnes contributed equally to this article. Author order was determined by mutual decision by both authors based on who handled the submission and paper revision process.

The authors declare no conflict of interest.

however, as asymptomatic infections are common, many CT infections go unnoticed without routine screening (2). Nucleic acid amplification tests (NAATs) like real-time PCR and transcription-mediated amplification (TMA) have been the most frequent NAATs used in clinical and public health laboratories, often with concomitant detection of *Neisseria gonorrhoeae* (or gonococcus [GC]) (3, 4). Like any molecular method (5), NAATs for CT/GC are designed to target conserved genomic regions, but mutations can arise over time with microbial evolution. If sequence mismatches occur within the NAAT target site, reduced sensitivity or detection failure could ensue (5–10).

False negative CT results occurred on a commonly used commercial NAAT for CT/GC testing, the Hologic Aptima Combo 2 (AC2) assay on the Panther instrument (Hologic Inc., San Diego, CA, USA) (11–17). The culprit termed the Finnish new variant of CT (FI-nvCT) harbored a 23S rRNA C1515T mutation, corresponding to the target site of the acridinium probe used for the CT detection. The emergence of FI-nvCT causing false negative results prompted surveillance in many countries (11–17). This variant along with other diagnostic escape mutations (i.e., C1514T, A1518G, C1522T, G1523A, and G1526A) were identified in Finland, and other Nordic countries (i.e., Sweden, Norway, and Denmark), England, Japan, and recently the United States (11–17). This study reports surveillance for CT diagnostic escape mutants on the AC2 assays in Canada from June 2019 to August 2021, prior to the introduction of a new reformulated assay with a novel CT probe.

Swab or urine specimens in Aptima Specimen Transport Media were submitted to the National Microbiology Laboratory (NML, Winnipeg, MB) from participating Canadian provincial public health laboratories across five Canadian provinces (i.e., Alberta [AB], British Columbia [BC], Ontario [ON], Quebec [QC], and Saskatchewan [SK]). As per previous recommendations (12), eligible samples were those with equivocal results on the AC2 CT/GC assay on the Panther instrument with relative light unit (RLU) values between 25 and 99, that were positive (>100 RLU) following reflex testing on the Aptima Chlamydia Trachomatis (ACT) assay (which detects an alternative CT-specific target, the 16S rRNA). All eligible samples were queued for 23S rRNA gene sequencing targeted by the AC2 assay. For each specimen, 500 µL were processed as per manufacturer instructions using the DNA and Viral NA Large Volume Kit (Roche Diagnostics, cat# 06374891001) on a MagNAPure 96 instrument (Roche Diagnostics, Laval, QC), and nucleic acids were eluted in 50 µL. For 23S rRNA gene sequencing, a nested RT-PCR protocol was provided by Dr. Damon Getman (Scientific Director), Hologic Inc. The first round RT-PCR was performed with 5 µL of template in 25 µL reactions using primers and concentrations defined in Table S1, and the buffer and enzymes from the Invitrogen SuperScript III One-Step RT-PCR System with Platinum Taq (Invitrogen, Cat. # 12574026). Amplification conditions were as follows: reverse transcription (55°C, 30 min), initial denaturation (94°C, 2 min), followed by 40 cycles of denaturation (94°C, 15 s), annealing (56.5°C, 30 s), and extension (68°C, 1 min). After final elongation (68°C, 5 min), amplicons were held at 4°C until use. The second round of PCR was performed in 25 µL reactions consisting of primers (Table S1), 1 µL of template from the first-round reaction, and buffers and enzymes from the OneStep Ahead RT-PCR kit (Qiagen Inc., Cat. # 220211). Cycling conditions were as follows: initial denaturation (95°C, 5 min), followed by 40 cycles of denaturation (95°C, 10 s), annealing (56.5°C, 10 s), and extension (72°C, 10 s). Final elongation (72°C, 2 min) was followed by holding at 4°C until use. The resulting amplicons were visualized using the QIAxcel Advanced system (Qiagen, Cat. # 9001941), according to manufacturer instructions. Sanger sequencing was performed by the NML DNA Genomics Core Facility. Specimens yielding sequencing results were analyzed using the Vector NTI 11.5.3 and MEGA6 (18), with alignments to reference CT serovars D, E, and J using the National Center for Biotechnology Information (NCBI) Genbank database accession numbers CP007131.1, HE601870.1, and CP017741.1, respectively. Additionally, nucleic acids were subjected to an in-house real-time quantitative PCR (qPCR) targeting the Chlamydia trachomatis cryptic plasmid as previously described (19) and crossing point (Cp) values were recorded.

Of 313 samples submitted for sequencing, sequencing failures occurred in 146 (46.6%) (Table 1). This high proportion of specimens with low sequencing efficiency has been reported by others for AC2 equivocal specimens and is likely due to specimens with low bacterial loads falling below or near the limit of detection of the sequencing reactions (15, 20). Supporting this hypothesis, 93 (63.7%) of the 146 specimens with sequencing failures were negative for the cryptic plasmid (CP) target, with Cp values ≥40 using an in-house real-time qPCR. The Cp values of the diagnostic escape variants were significantly lower ($P < 0.0001$) than those of strains with wild-type sequences, suggesting higher bacterial loads were present (Fig. S2). In total, 190 samples were CP positive, 115 were negative, and 8 were NSQ as tested by in-house real-time quantitative PCR.

Of the 167 successfully sequenced specimens, 152 (91.0%) contained wild-type 23S rRNA sequences. These sequences most closely resembled serovars D (5 [3.3%]), E (47 [31.0%]), and J (100 [65.8%]) within the target region (Fig. S1; Table 1). The remaining 15 sequences harbored mutations known to cause diagnostic escape with the AC2 assay (11–17): C1514T ($n = 8$), G1526A ($n = 5$), and one each with G1523A or C1522T (Fig. S1; Table 1). G1523A, C1514T, and C1522T were previously reported by others (13–15, 21), but the G1526A mutation was more recently described in the United States (11). No sequences corresponded to the other mutations causing detection failures for the AC2 assay, e.g., C1515T (16, 17) or A1518G (11).

While the proportion of CT diagnostic escape variants during the study period is estimated to be low (<0.01%), this may be underestimated due to the inherent biases of specimen selection (i.e., focus on equivocal samples) (11, 12), the voluntary submissions from participating laboratories, and the sensitivity limitations of sequencing methods. Parallel testing of all specimens with the AC2 and ACT assays (or between other NAATs) could be a solution to identify additional CT variant cases; however, this strategy was not possible with the high volumes of CT/GC testing occurring in clinical and public health laboratories. Regardless, the low proportion of CT variants seen in this study was consistent with other countries using a similar surveillance strategy (11–17). For example, the prevalence of AC2 diagnostic escape mutants (i.e., C1514T and G1523A) was <0.003% in England (13). Despite the low prevalence of diagnostic escape mutants observed in this and other studies, there was an ongoing risk of further transmissions. As seen with the Swedish novel CT variant (SW-nvCT) with NAATs used at the time, unrecognized circulation of variants can lead to significant disease spread (5–7). Hologic Inc. recently redesigned the AC2 assay to include an additional CT detection probe falling outside the region affected by the recent 23S rRNA diagnostic escape variants (11, 20, 21), which is a common strategy to mitigate NAAT target site mutations (5, 22–25). Following retesting with the reformulated AC2 assay, all specimens with CT variants from this study were found to be positive (Table 2).

**TABLE 1** Summary of 23S rRNA sequencing and variants detected

| Description | Province | | | | | |
|---|---|---|---|---|---|---|
| | AB | BC | ON | QC | SK | Total |
| C1515T (FI-nvCT) | 0 | 0 | 0 | 0 | 0 | 0 |
| C1514T | 5 | 0 | 3 | 0 | 0 | 8 |
| C1522T | 1 | 0 | 0 | 0 | 0 | 1 |
| G1523A | 1 | 0 | 0 | 0 | 0 | 1 |
| G1526A | 5 | 0 | 0 | 0 | 0 | 5 |
| WT (serovar D)[a] | 0 | 0 | 0 | 5 | 0 | 5 |
| WT (serovar E)[a] | 6 | 10 | 8 | 18 | 5 | 47 |
| WT (serovar J)[a] | 39 | 8 | 28 | 20 | 5 | 100 |
| Sequencing failed | 49 | 6 | 35 | 33 | 23 | 146 |
| Total | 106 | 24 | 74 | 76 | 33 | 313 |

[a]Wild-type (WT) sequences for CT most closely resembling serovars D, E, and J in the 23S rRNA target region were derived from Genbank accession numbers CP007131.1, HE601870.1, and CP017741.1, respectively.

**TABLE 2** Analytical characteristics of the 15 diagnostic escape variants detected in this study

| NML identifier | Results | | | | |
|---|---|---|---|---|---|
| | 23S rRNA mutation | AC2 (RLU)[a] | ACT (RLU)[a] | qPCR (Cp)[a] | Reformulated AC2 (RLU)[a] |
| CLM20-1137 | C1514T | 70 | 7258 | 27.28 | 556 |
| CLM20-1139 | C1514T | 82 | 7081 | 20.04 | 393 |
| CLM21-1417 | C1514T | 38 | 6641 | 23.62 | 457 |
| CLM21-1442 | C1514T | 33 | 6936 | 27.63 | 417 |
| CLM19-5277 | C1514T | 53 | 6109 | 29.29 | 360 |
| CLM19-5283 | C1514T | 26 | 6078 | 30.55 | 401 |
| CLM19-5517 | C1514T | 90 | 7600 | 20.87 | 396 |
| CLM21-1455 | C1514T | 47 | 9112 | 27.64 | 405 |
| CLM22-0087 | C1522T | 26 | 7013 | 28.39 | 423 |
| CLM19-5278 | G1523A | 88 | 8110 | 33.73 | 594 |
| CLM21-1393 | G1526A | 32 | 6733 | 29.46 | 533 |
| CLM21-1402 | G1526A | 53 | 6327 | 25.60 | 591 |
| CLM21-1405 | G1526A | 53 | 6903 | 24.35 | 549 |
| CLM22-0090 | G1526A | 54 | 6749 | 30.91 | 650 |
| CLM22-0089 | G1526A | 38 | 6799 | 30.35 | 649 |

[a]The AC2, ACT, and reformulated AC2 assay were performed on the Panther instrument, where the qPCR represents an in-house real-time PCR targeting the cryptic plasmid. While eligibility criteria were followed for testing, these data were not submitted by the participating laboratory.

Overall, the presence of mutations in the AC2 target region emphasizes the importance of conducting regular genomic surveillance to monitor the evolution and diversity of Chlamydia Trachomatis serovars circulating within the country.

## ACKNOWLEDGMENTS

The authors would like to thank all the members of the Canadian Public Health Laboratory Network (CPHLN) Chlamydia Variant Working Group, Dr. Damon Getman (Scientific Director, Hologic Inc.), and laboratory support staff for their contributions to the study.

Designed the study: AS2; Performed the experiments: J.B., F.R., S.G.; Analysed the data: F.R., J.B., AS1; Wrote the first draft of the paper: F.R., J.B., AS2, J.J.L.

All authors contributed to the editing and finalization of the manuscript.

## AUTHOR AFFILIATIONS

[1]National Microbiology Laboratory Branch, Public Health Agency of Canada, Winnipeg, Manitoba, Canada
[2]Division of Microbiology, Department of Pathology and Laboratory Medicine, Nova Scotia Health, Halifax, Nova Scotia, Canada
[3]Departments of Pathology, Medicine (Infectious Diseases), and Microbiology and Immunology, Dalhousie University, Halifax, Nova Scotia, Canada
[4]Department of Medical Microbiology and Infectious Diseases, Faculty of Health Sciences, University of Manitoba, Winnipeg, Manitoba, Canada
[5]Centre for the AIDS Programme of Research in South Africa (CAPRISA), Durban, South Africa
[6]Department of Medical Microbiology, University of KwaZulu-Natal, Durban, South Africa

## AUTHOR ORCIDs

Felicia Roy  http://orcid.org/0009-0000-2568-6721
Jason J. LeBlanc  http://orcid.org/0000-0003-0593-0357
Alberto Severini  http://orcid.org/0000-0002-0844-2634

## AUTHOR CONTRIBUTIONS

Felicia Roy, Data curation, Methodology, Writing – original draft, Writing – review and editing | Jennifer Beirnes, Data curation, Methodology, Writing – original draft, Writing – review and editing | Jason J. LeBlanc, Writing – original draft, Writing – review and editing | Suzanne Gibbons, Methodology, Writing – review and editing | Aida Sivro, Writing – original draft, Writing – review and editing | Alberto Severini, Conceptualization, Writing – original draft, Writing – review and editing

## ADDITIONAL FILES

The following material is available online.

### Supplemental Material

**Supplemental table and figures (Spectrum02062-24-s0001.docx).** Table S1; Fig. S1 and S2.

### Open Peer Review

**PEER REVIEW HISTORY (review-history.pdf).** An accounting of the reviewer comments and feedback.

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
