## [Reviewer comments · Microbiology Spectrum]

Microbiology Spectrum

Surveillance for *Chlamydia trachomatis* variants escaping detection with the Aptima Combo 2 assay in Canada from 2019 to 2021

Felicia Roy, Jennifer Beirnes, Jason LeBlanc, Suzanne Gibbons, Aida Sivro, and Alberto Severini

Corresponding Author(s): Felicia Roy, Public Health Agency of Canada

Review Timeline:

Submission Date:	August 20, 2024
Editorial Decision:	November 1, 2024
Revision Received:	December 2, 2024
Accepted:	December 5, 2024

Editor: Mark Pandori

Reviewer(s): Disclosure of reviewer identity is with reference to reviewer comments included in decision letter(s). The following individuals involved in review of your submission have agreed to reveal their identity: Collette Bromhead (Reviewer #2); Jennifer Anne Danielewski (Reviewer #3)

Transaction Report:

DOI: <https://doi.org/10.1128/spectrum.02062-24>

Re: Spectrum02062-24 (Surveillance for Chlamydia trachomatis variants escaping detection with the Aptima Combo 2 assay in Canada from 2019 to 2021)

Dear Dr. Felicia Roy:

Thank you for the privilege of reviewing your work. Below you will find my comments, instructions from the Spectrum editorial office, and the reviewer comments.

Revision Guidelines

Sincerely,
Mark Pandori
Editor
Microbiology Spectrum

Reviewer #2 (Comments for the Author):

Thank you for the opportunity to review this nice succinct paper, the findings of which support a more routine genomic surveillance programme of *C. trachomatis* NAAT target sequence diversity and evolution. It has become clear since the emergence of the "Finnish mutant" and the earlier "Swedish mutant" that, unlike *N. gonorrhoeae*, *C. trachomatis* is making something of a habit of responding to evolutionary pressure by evolving its diagnostic genomic targets faster than its therapeutic

targets.

I have some questions and small suggestions for improvement:

Line 87: suggest you change "Clinical manifestations" to "Clinical sequelae". This is because pelvic inflammatory disease, ectopic pregnancy etc are relatively infrequent outcomes of CT infection compared to e.g. dysuria.

Line 103 to 105: I wouldn't describe your study as surveillance. Suggest you adjust your sentence describing the study to " this is a retrospective study of the frequency and types of CT diagnostic escape mutants of the AC2 assay in a cohort of equivocal samples collected between 2019-2021etc.

Lines 167-168: Suggest that, given the new AC2 assay is already in use in Canada, that the results of your study "support more regular genomic surveillance of the evolution and diversity among Chlamydia trachomatis serovars circulating in Canada." Or words to that effect.

Questions:

1. I have re-read Unemo et al 2019 (your reference 12) and see they recommend retesting equivocal / negative samples with RLU between 20-85, which is not consistent with your statements on lines 109 to 111. Can you explain this difference in your sample selection? Was it to do with sample availability?
2. Unemo et al 2019 also recommend whole genome sequencing, I am curious as to why this was not your method of choice, nor mentioned as a future tool for genomic surveillance?
3. It might just be my reading of the section but I'm unsure why the last sentence of the paragraph from line 135 to 142 makes a point that Ct values were lower in diagnostic escape variants compared to wildtype strains,- although earlier you made a point of reporting how many sequencing failures there were due to equivocal samples having low bacterial loads. Can you clarify for me? (btw yes I know about the inverse relationship between Ct values and sample load)
4. Line 151, yes your sample selection was biased. You could have included negative samples from symptomatic people as well as positive samples to reduce bias.
5. Why is there no information on the N. gonorrhoeae status of the samples in the study?

Reviewer #3 (Comments for the Author):

Accept with some modifications.

Please see attached document for reviewer comments.

Review of Roy et al 2024: Surveillance for *C. trachomatis* variants escaping detection with the Aptima Combo 2 assay in Canada from 2019 to 2021

Thank you for the opportunity to review this nice succinct paper, the findings of which support a more routine genomic surveillance programme of *C. trachomatis* NAAT target sequence diversity and evolution. It has become clear since the emergence of the “Finnish mutant” and the earlier “Swedish mutant” that, unlike *N. gonorrhoeae*, *C. trachomatis* is making something of a habit of responding to evolutionary pressure by evolving its diagnostic genomic targets faster than its therapeutic targets.

I have some questions and small suggestions for improvement:

Line 87: suggest you change “Clinical manifestations” to “Clinical sequelae”. This is because pelvic inflammatory disease, ectopic pregnancy etc are relatively infrequent outcomes of CT infection compared to e.g. dysuria.

Line 103 to 105: I wouldn’t describe your study as surveillance. Suggest you adjust your sentence describing the study to “ this is a retrospective study of the frequency and types of CT diagnostic escape mutants of the AC2 assay in a cohort of equivocal samples collected between 2019-2021etc.

Lines 167-168: Suggest that, given the new AC2 assay is already in use in Canada, that the results of your study “support more regular genomic surveillance of the evolution and diversity among *Chlamydia trachomatis* serovars circulating in Canada.” Or words to that effect.

Questions:

1. I have re-read Unemo et al 2019 (your reference 12) and see they recommend retesting equivocal / negative samples with RLU between 20-85, which is not consistent with your statements on lines 109 to 111. Can you explain this difference in your sample selection? Was it to do with sample availability?
2. Unemo et al 2019 also recommend whole genome sequencing, I am curious as to why this was not your method of choice, nor mentioned as a future tool for genomic surveillance?
3. It might just be my reading of the section but I’m unsure why the last sentence of the paragraph from line 135 to 142 makes a point that Ct values were lower in diagnostic escape variants compared to wildtype strains,— although earlier you made a point of reporting how many sequencing failures there were due to equivocal samples having low bacterial loads. Can you clarify for me? (btw yes I know about the inverse relationship between Ct values and sample load)
4. Line 151, yes your sample selection was biased. You could have included negative samples from symptomatic people as well as positive samples to reduce bias.

5. Why is there no information on the *N. gonorrhoeae* status of the samples in the study?

Surveillance for Chlamydia trachomatis variants escaping detection with the Aptima Combo 2 assay in Canada from 2019 to 2021

Felicia Roy, Jennifer Beirnes, Jason LeBlanc, Suzanne Gibbons, Aida Sivro, Alberto Severini

Reviewer comments:

The manuscript is both succinct and clearly presented. The importance of investigating the rates of false negatives for routine diagnostic assays cannot be overstated in the context of the diagnostic setting. Publication of studies demonstrating the presence of escape variants and alternative diagnostic assays for implementation, or for identifying rates of false negatives in other populations, provide diagnostic laboratories the ability to implement accurate workflows.

General comments:

Overall, the paper is well written and clearly presented. I expect that the authors performed the sanger sequencing using an RNA based protocol as the Aptima Combo assay is RNA based (rather than amplifying the 23S rRNA gene directly)? I was unsure as to why there was a need to perform nested PCR with a different enzyme, and why the product of the first round PCR could not be directly sequenced. Was it to increase amplicon abundance?

Additionally, I understand the focus of the paper is to report on the frequency of escape mutants in the sampled population, determine the mutation types present and offer an alternative assay for detecting these escape mutants. The authors have presented a great study in this regard.

I think what this paper lacks is some discussion on the QC points used in both methods and discussion on the overall results. For example, how did the QC points correlate with sequence outcomes? This is touched on for a portion of sequence failure but there is no discussion from the authors on the proportion of *suspected false negatives* that were actually *negatives*. That is, *not detected throughout or outside the limit of detection* for some or all of the assays used. The overall frequency of “*negative/not detected*” from the 313 *suspected false negatives*, that were not escape mutants is also relevant and provides important information for diagnostic services. This information warrants some discussion and could be presented in a flow diagram or supplementary table.

Specific Comments:

Line 52 – Suggest rewording of this sentence. For the 15 samples which had one of the three variants noted, are you able to provide a figure (%) indicating the proportion of the samples suspected to be *false negatives* which were found to have escape variations? This helps with context by stating the number of false negatives/equivocal due to low load infections vs load of escaped mutants. Important as you report escape mutants having higher loads and this has an impact on patient outcome and treatment.

Line 111 – Can the authors outline the criteria used in Canada for reflex testing in this section.

Line 113 – Can the authors include the number of samples, and proportion of the total tested, that were identified as potential false negatives in this sentence.

Line 116-118 – Although the assay used is referenced, additional clarification is recommended here. For example, inclusion of the full name (*Chlamydia trachomatis* cryptic plasmid) and amplicon length.

In addition, stating the rationale for using this assay. Generally, these are used to confirm nucleic acid extraction was successful, which is why a non-target human housekeeping gene is often used. Explanation of why a *Chlamydia trachomatis* plasmid target was chosen should be included. Was this a confirmation of *C trachomatis* in the sample prior to performing PCR protocol for sequencing? If so, as the “values” are not presented (with the exception of 15 samples in Table 2) the QC parameters or number of samples detected/not-detected should be included here for all samples, not just in association with the samples which failed sequencing (line 139-140 and reference to Figure S2). See also comments for line 130 below.

Line 120 – Table S1 Please clarify which primer pair was the outer/first and which were the inner/second used for the PCR. Also please add the amplicon length generated by the sequencing primers.

Line 130- Given the high number of samples which failed Sanger sequencing (line 135), it prompts the question. What were your quality criteria (QC) for samples sent for sequencing? These are not specified for in-house qPCR (Cp exclusion value) or for the QIAxcel in the methods. If samples were not detected for the in-house assay (DNA), were they still all used for the nested RT-PCR protocol (RNA)? This is not stated but seems that they were from Table 1. If so, why (rationale)? Did samples have a visible band and/or were these quantitated post purification prior to sequencing? These are commonly used QC points prior to sequencing that are used to exclude samples with insufficient amplicon for sequencing.

December 02, 2024

To:

Mark Pandori
Editor
Microbiology Spectrum

RE: Response to reviewers for manuscript: "Surveillance for Chlamydia trachomatis variants escaping detection with the Aptima Combo 2 assay in Canada from 2019 to 2021" (Spectrum02062-24)

Thank you for considering our manuscript for publication in *Microbiology Spectrum* journal. We would also like to thank the reviewers for their input and feedback. Please find the responses to reviewers comments below.

Reviewer 2:

Line 87: suggest you change "Clinical manifestations" to "Clinical sequelae". This is because pelvic inflammatory disease, ectopic pregnancy etc are relatively infrequent outcomes of CT infection compared to e.g. dysuria.

Thank you for the suggestion. We have adjusted the language as suggested.

Line 103 to 105: I wouldn't describe your study as surveillance. Suggest you adjust your sentence describing the study to " this is a retrospective study of the frequency and types of CT diagnostic escape mutants of the AC2 assay in a cohort of equivocal samples collected between 2019-2021etc.

Thank you for your comment. We would like to clarify that this study was not retrospective. Testing was conducted in real time as part of routine diagnostic surveillance workflows at the National Microbiology Laboratory's Sexually Transmitted Infections Reference Laboratory.

Lines 167-168: Suggest that, given the new AC2 assay is already in use in Canada, that the results of your study "support more regular genomic surveillance of the evolution and diversity among *Chlamydia trachomatis* serovars circulating in Canada." Or words to that effect.

Thank you for the suggestion. We have adjusted the sentence as follows: "Overall, the presence of mutations in the AC2 target region emphasizes the importance of conducting regular genomic surveillance to monitor for the evolution and diversity of Chlamydia Trachomatis serovars circulating within the country."

Questions:

1. I have re-read Unemo et al 2019 (your reference 12) and see they recommend retesting equivocal / negative samples with RLU between 20-85, which is not consistent with your statements on lines 109 to 111. Can you explain this difference in your sample selection? Was it to do with sample availability?

Thank you for the thoughtful comment. The schema to retest equivocal samples between 25 to 99 RLU was based on the published equivocal RLU range (25 to < 100) from the Aptima Combo 2 (AC2) Test Interpretation Kit Insert (Version # 502487 Rev. 004) and in consultations with the Canadian Public Health Laboratory Network (CPHLN) Chlamydia Variant Working Group.

2. Unemo et al 2019 also recommend whole genome sequencing, I am curious as to why this was not your method of choice, nor mentioned as a future tool for genomic surveillance?

Thank you for the comment. At the time of the study we did not have access to whole genome sequencing protocols and did not have the capacity to provide WGS. The feasibility was not practical from a diagnostic/reference testing stand point as we were providing testing in real-time to external stakeholders and were adhering to established turn around times.

3. It might just be my reading of the section but I'm unsure why the last sentence of the paragraph from line 135 to 142 makes a point that Ct values were lower in diagnostic escape variants compared to wildtype strains, – although earlier you made a point of reporting how many sequencing failures there were due to equivocal samples having low bacterial loads. Can you clarify for me? (btw yes I know about the inverse relationship between Ct values and sample load)

Thank you for your comment. The Ct values of the diagnostic escape variants were lower, meaning the real time PCR positivity crossing threshold was crossed sooner thereby meaning the sample was more positive suggesting higher bacterial loads were present in those samples. In contrast, wild type specimens falling in the equivocal / negative Panther RLU range are likely to fail sequencing because they are specimens with low bacterial loads to begin with and falsely suspected of being Finnish Variant specimens.

Recall that eligible samples were those with equivocal results on the AC2 CT/GC assay on the Panther instrument with relative light unit (RLU) values between 25 and 99, that were positive (>100 RLU) following reflex testing on the Aptima Chlamydia Trachomatis Assay (ACT) assay (which detects an alternative CT-specific target, the 16S rRNA).

It is proposed that wild type specimens that met this criteria were simply specimens with low bacterial load and contributed to the high rate of sequence failures.

4. Line 151, yes your sample selection was biased. You could have included negative samples from symptomatic people as well as positive samples to reduce bias.

Thank you for the comment. We have included the limitations of the study in the manuscript and as explained previously, this was not a retrospective study design but a report on a diagnostic surveillance program that took place during 2019 to 2021.

The routine diagnostic surveillance workflows at the National Microbiology Laboratory's Sexually Transmitted Infections Reference Laboratory include receiving specimens positive for Chlamydia trachomatis that meet criteria for LGV serovar differentiation. Recall that eligible samples were those with equivocal results on the AC2 CT/GC assay on the Panther instrument with relative light unit (RLU) values between 25 and 99, that

were positive (>100 RLU) following reflex testing on the Aptima Chlamydia Trachomatis Assay (ACT) assay (which detects an alternative CT-specific target, the 16S rRNA).

5. Why is there no information on the *N. gonorrhoeae* status of the samples in the study?

Thank you for the comment. Information of NG status was not provided by sending laboratories. This surveillance was specific for mutations impacting *C. trachomatis* molecular target detection on the Hologic Panther AC2 assay that had been reported in several countries, raising concerns of potential false negative results. This surveillance showed *C. trachomatis* target detection failures in specimens submitted for *C. trachomatis* testing in Canadian laboratories from 2019 to 2021.

Reviewer 3:

Line 52 – Suggest rewording of this sentence. For the 15 samples which had one of the three variants noted, are you able to provide a figure (%) indicating the proportion of the samples suspected to be *false negatives* which were found to have escape variations? This helps with context by stating the number of false negatives/equivocal due to low load infections vs load of escaped mutants. Important as you report escape mutants having higher loads and this has an impact on patient outcome and treatment.

Thank you for the thoughtful suggestion. We have adjusted the sentence as follows: Sequencing revealed 15 (4.8%) diagnostic escape variants which were carrying either C1514T, G1523A, or G1526A mutations.

Line 111 – Can the authors outline the criteria used in Canada for reflex testing in this section.

Thank you for your thoughtful comment. The criteria used in Canada for reflex testing was determined from published references (noted in the manuscript as reference # 12 - Unemo et al.) and with the Canadian Public Health Laboratory Network (CPHLN) Chlamydia Variant Working Group in addition to Dr. Damon Getman (Scientific Director, Hologic.)

Line 113 – Can the authors include the number of samples, and proportion of the total tested, that were identified as potential false negatives in this sentence.

Thank you for your thoughtful comment. The proportion of the total samples tested is outlined in lines 134 to 141.

Line 116-118 – Although the assay used is referenced, additional clarification is recommended here. For example, inclusion of the full name (*Chlamydia trachomatis* cryptic plasmid) and amplicon length.

In addition, stating the rationale for using this assay. Generally, these are used to confirm nucleic acid extraction was successful, which is why a non-target human housekeeping gene is often used. Explanation of why a *Chlamydia trachomatis* plasmid target was chosen should be included. Was this a confirmation of *C trachomatis* in the sample prior to performing PCR protocol for sequencing? If so, as the “values” are not presented (with the exception of 15 samples in Table 2) the QC parameters or number of samples detected/not-detected should be included here for all samples, not just in association with the samples which failed sequencing (line139-140 and reference to Figure S2). See also comments for line 130 below.

Thank you for the thoughtful comment. We have adjusted the sentence as follows: “For each specimen, 500 µL were processed as per manufacturer instructions using the DNA and Viral NA Large Volume Kit (Roche Diagnostics, cat# 06374891001) on a MagNAPure 96 instrument (Roche Diagnostics, Laval, QC), and nucleic acids were eluted in 50 µL. Nucleic acids were subjected to an in-house real-time quantitative PCR (qPCR) targeting the *Chlamydia trachomatis* cryptic plasmid as previously described [18] and crossing point (Cp) values were recorded.”

The rationale for using the in-house real-time quantitative PCR (qPCR) targeting the *Chlamydia trachomatis* cryptic plasmid was to confirm the presence of *Chlamydia* and to estimate the bacterial load. This in-house quantitative assay is a quadra-plex assay with probes to detect LGV, non-LGV, cryptic plasmid, and Rnase P.

Line 120 – Table S1 Please clarify which primer pair was the outer/first and which were the inner/second used for the PCR. Also please add the amplicon length generated by the sequencing primers.

Thank you for the suggestion. We have added the required information to the table as suggested.

Line 130- Given the high number of samples which failed Sanger sequencing (line 135), it prompts the question. What were your quality criteria (QC) for samples sent for sequencing? These are not specified for in-house qPCR (Cp exclusion value) or for the QIAxcel in the methods. If samples were not detected for the in-house assay (DNA), were they still all used for the nested RT-PCR protocol (RNA)? This is not stated but seems that they were from Table 1. If so, why (rationale)? Did samples have a visible band and/or were these quantitated post purification prior to sequencing? These are commonly used QC points prior to sequencing that are used to exclude samples with insufficient amplicon for sequencing.

What were your quality criteria (QC) for samples sent for sequencing?

Thank you for the thoughtful comment. Testing was conducted in real time as part of routine diagnostic surveillance workflows at the National Microbiology Laboratory’s Sexually Transmitted Infections Reference Laboratory, quality criteria for samples sent for Sanger sequencing were generous as the real-time qPCR is less sensitive than the ACT and AC2 assays as it is not a diagnostic assay, it is to differentiate LGV vs Non-LGV serovars on confirmed *Chlamydia Trachomatis* positive samples. This high proportion of

specimens with low sequencing efficiency has been reported by others for AC2 equivocal specimens, and is likely due to specimens with low bacterial loads falling below or near the limit of detection of the sequencing reactions [15,20], not failed quality criteria.

If samples were not detected for the in-house assay (DNA), were they still all used for the nested RT-PCR protocol (RNA)?

That is correct. Irrespective of the qPCR result all samples were still tested. This has been clarified in the manuscript.

If so, why (rational)?

The rationale for using the in-house real-time quantitative PCR (qPCR) targeting the *Chlamydia trachomatis* cryptic plasmid (DNA) was to confirm the presence of *Chlamydia* and to estimate the bacterial load. As the in house assay is less sensitive than the commercial AC2/ACT assays we did not want to miss any potential low bacterial load samples.

Did samples have a visible band and/or were these quantitated post purification prior to sequencing?

No, all samples were submitted for sequencing, irrespective of the presence of the visible band. This has been clarified in the manuscript.

Please let us should you require additional clarifications and edits.

Sincerely,

Felicia Roy

Felicia Roy

Re: Spectrum02062-24R1 (Surveillance for Chlamydia trachomatis variants escaping detection with the Aptima Combo 2 assay in Canada from 2019 to 2021)

Dear Dr. Felicia Roy:

Your manuscript has been accepted, and I am forwarding it to the ASM production staff for publication. Your paper will first be checked to make sure all elements meet the technical requirements. ASM staff will contact you if anything needs to be revised before copyediting and production can begin. Otherwise, you will be notified when your proofs are ready to be viewed.

Sincerely,
Mark Pandori
Editor
Microbiology Spectrum